# Separability and geometry of object manifolds in deep neural networks

Uri Cohen [1,6], SueYeon Chung [2,3,4,6], Daniel D. Lee[5] & Haim Sompolinsky[1,2]*

Stimuli are represented in the brain by the collective population responses of sensory neurons, and an object presented under varying conditions gives rise to a collection of neural population responses called an 'object manifold'. Changes in the object representation along a hierarchical sensory system are associated with changes in the geometry of those manifolds, and recent theoretical progress connects this geometry with 'classification capacity', a quantitative measure of the ability to support object classification. Deep neural networks trained on object classification tasks are a natural testbed for the applicability of this relation. We show how classification capacity improves along the hierarchies of deep neural networks with different architectures. We demonstrate that changes in the geometry of the associated object manifolds underlie this improved capacity, and shed light on the functional roles different levels in the hierarchy play to achieve it, through orchestrated reduction of manifolds' radius, dimensionality and inter-manifold correlations.

[1] Edmond and Lily Safra Center for Brain Sciences, Hebrew University of Jerusalem, Jerusalem, Israel. [2] Center for Brain Science, Harvard University, Cambridge, MA, USA. [3] Department of Brain and Cognitive Sciences, Massachusetts Institute of Technology, Cambridge, MA, USA. [4] Center for Theoretical Neuroscience, Columbia University, New York, NY, USA. [5] Department of Electrical and Computer Engineering, Cornell Tech, New York, NY, USA. [6]These authors contributed equally: Uri Cohen, SueYeon Chung. *email: haim@fiz.huji.ac.il

The visual hierarchy of the brain has a remarkable ability to identify objects despite differences in appearance due to changes in variables such as position, illumination, and background[1]. Recent research in machine learning has shown that deep convolutional neural networks (DCNNs)[2] can perform invariant object categorization with almost human-level accuracy[3], and that their network representations are similar to the brain's[4–6]. DCNNs are therefore very important as models of visual hierarchy[7–9], though understanding their operational capabilities and design principles remain a significant challenge[10].

In a visual hierarchy, the neuronal population response to stimuli belonging to the same object defines an object manifold. The brain's ability to discriminate between objects can be mapped to the separability of object manifolds by a simple, biologically plausible readout, modeled as a linear hyperplane[11]. It has been hypothesized that the visual hierarchy untangles object manifolds to transform inseparable object manifolds into linearly separable ones, as illustrated in Fig. 1. This intuition underlies a number of studies on object representations in the brain[1,12–15], and in deep artificial neural networks[16–19]. As separability of manifolds depends on numerous factors—numbers of neurons and manifolds, sizes and shapes of the manifolds, target labels, among others—it has been difficult to elucidate which specific properties of the representation truly contribute to untangling. Quantifying the changes in neural representations between different sensory stages has been a focus of research in computational neuroscience[10,20–22]. Canonical correlation analysis (CCA) has recently been proposed to compare the representations in hierarchical deep networks[23,24]. Another approach, representational similarity analysis (RSA), uses similarity matrices to determine which stimuli are more correlated within neural data and in network representations[4,7,25]. Others have considered various measures such as curvature[15] and dimensionality to capture the structure within neural representations[26–32]; but it is unclear how these measures are related to task performance such as classification accuracy. Conversely, others have explored functional aspects by using different layer representations for transfer learning[33], or object classification[31], but it is unclear why performance improves or deteriorates. Other attempts to characterize representations have focused primarily on single-neuron properties in relation to object invariance[17,34,35].

In this work, we apply the theory of linear separability of manifolds[36] to analytically demonstrate that separability depends on three measurable quantities, manifold dimension and extent, and inter-manifold correlation. This result provides a powerful new tool to elucidate how changes in object representations within deep networks underlie the emergence of manifold untangling and separability. We show how to apply these measures to reveal key changes in manifold geometries in several representative DCNNs currently used for object categorization. Thus, our analysis provides novel insight into the functional role of sensory hierarchies in the processing of categorical information.

## Results

**Geometrical framework.** We start our analysis by defining the separability of objects on the basis of their representations in a given neuronal populations. We denote a collection of objects as linearly separable (or simply separable) if they can be classified into two desired classes by a hyperplane in the state space of the population (Fig. 1). Specifically, we consider a layer consisting of $N$ neurons representing $P$ object manifolds; the system load is defined by the ratio $\alpha = P/N$[37]. We ask whether these manifolds can be separated by a hyperplane. In the regime where $P$ and $N$ are large, our theory shows the existence of a critical load value $\alpha_c$,

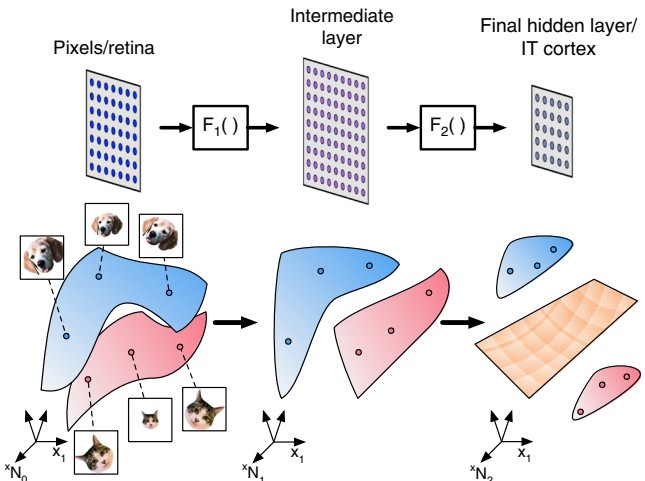

**Fig. 1 Changes in the geometry of object manifolds as they are transformed in a deep neural network.** Illustration of three layers in a visual hierarchy where the population response of the first layer is mapped into intermediate layer by $F_1$ and into the last layer by $F_2$ (top). The transformation of per-stimuli responses is associated with changes in the geometry of the object manifold, the collection of responses to stimuli of the same object (colored blue for a 'dog' manifold and pink for a 'cat' manifold). Changes in geometry may result in transforming object manifolds which are not linearly separable (in the first and intermediate layers) into separable ones in the last layer (separating hyperplane, colored orange).

called manifold classification capacity, such that when $P < \alpha_c N$ object manifolds are separable with high probability, whereas if $P > \alpha_c N$ the manifolds are inseparable with high probability. That is, when assigning random binary labels to the $P$ manifolds, the probability of successful linear classification of the objects decreases sharply from 1 below capacity to 0 above it. Intuitively, this capacity serves as a measure of the linearly decodable information per neuron about object identity.

Next we study the role of manifolds' geometry on their classification capacity. It is very useful to consider several limiting cases. The largest possible value of $\alpha_c$ is 2[36–38] and is achieved when manifolds are just single points, i.e., fully invariant object representations. For a lower bound, we consider unstructured point-cloud manifolds, in which a set of $M \cdot P$ points is randomly grouped into $P$ manifolds. Since the points of each manifold do not share any special geometric features, the capacity is equal to that of $M \cdot P$ points[37,39] so the system is separable while $M \cdot P < 2N$, or equivalently $\alpha_c = 2/M$. Thus, the capacity of structured point-cloud manifolds consisting of $M$ points per manifold obeys $\frac{2}{M} \leq \alpha_c \leq 2$. Another limit of interest in unbounded smooth manifolds, each filling a $D$-dimensional linear subspace with randomly oriented axes, in which case $\alpha_c = 1/(D + 1/2)$[36].

Since real-world manifolds span high-dimensional subspaces and consist of a large or infinite number of points, a substantial capacity implies constraints on both the manifold dimensionality and extent. Realistic manifolds are expected to show inhomogeneous variations along different manifold axes, raising the question of how to assess their effective dimensionality and extent. Our theory provides a precise definition of the relevant manifold dimension and radius, denoted $D_M$ and $R_M$, that underlie their separability.

These quantities are derived from the structure of the hyperplane separating the manifolds when $P/N$ is close to capacity. When linearly classifying points, the structure of the separating hyperplane is determined by a set of input vectors

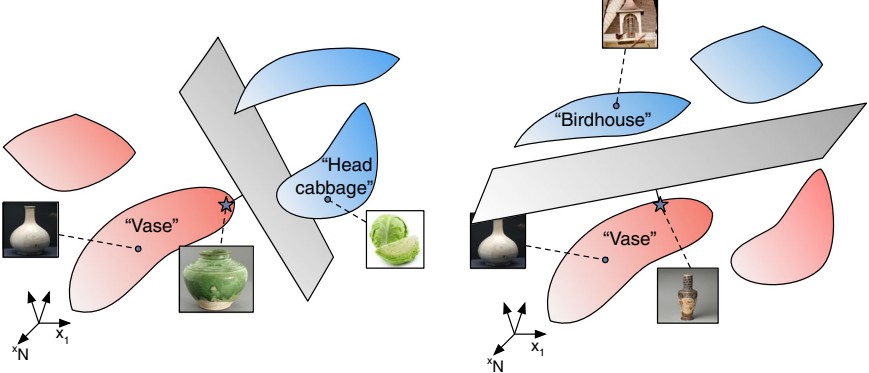

**Fig. 2 Anchor points in an object manifold.** Anchor points on manifolds define the optimal hyperplane separating a binary dichotomy of manifolds. This figure shows one realization of an anchor point on the 'vase' manifold (star) that separates it from the 'head cabbage' object manifold. Another realization illustrates a different anchor point on the 'vase' manifold when it is classified against the 'birdhouse' object manifold. The statistics of the distribution of these anchor points define the relevant geometrical manifold radius and manifold dimension of the 'vase' manifold that characterizes its linear classification properties (see Methods).

known as support vectors[40], namely, the weight vector normal to the separating plane is a linear combination of these vectors. As shown in ref. [36], this concept can be generalized to the case of manifolds, where the weight vector normal to their separating plane is a linear combination of anchor points. Each manifold contributes (at most) a single anchor point, which is a point residing in the manifold or in its convex hull. These points uniquely define the separating plane, thus anchoring it. The identity of the anchor points depends not only on the manifolds' shape but also on their location or orientation in state space as well as the particular choice of random labeling (Fig. 2). Thus, for a given fixed manifold, as the location and labeling of the other manifolds are varied, the manifold's anchor point will change, thereby generating a statistical distribution of anchor points for a manifold embedded in a statistical ensemble of other manifolds. The manifold's effective radius $R_M$ is the total variance of its anchor points normalized by the average distance between the manifold centers. Its effective dimension $D_M$ is the spread of the anchor points along the different manifold axes.

Our theory has shown that for manifolds occupying $D \gg 1$ dimensions (as in most cases of interest), $R_M$ and $D_M$ determine the classification capacity; in fact the capacity is similar to that of balls with radius and dimensions equal to $R_M$ and $D_M$, respectively (see Methods). Furthermore, using statistical mechanical mean-field techniques, we derive algorithms for measuring the capacity, $R_M$ and $D_M$ for manifolds given by either empirical data samples or from parametric generative models[36,41]. This theory assumed that the position and orientation of different manifolds are uncorrelated. Here we extend the theory and apply it to realistic manifolds with substantial inter-manifold correlations.

In the following, we apply this theory and algorithms to study how stages of deep networks transform object manifolds, illuminating the effect of architectural building blocks and non-linear operations on shaping manifold geometry and their correlations, and demonstrating their role in the enhancement of object classification capacity.

**Learning enhances manifold separability across layers.** We consider DCNNs trained for object recognition tasks on large labeled data-set, ImageNet[42]. Several state-of-the-art networks, such as AlexNet[43] and VGG-16[44], share similar computational building blocks consisting of alternating layers of linear convolutions, point-wise ReLU nonlinearities and max pooling, followed by several fully connected layers (Fig. 3a–c).

In each network, we measure classification capacity and geometry of point-cloud manifolds consisting of high scoring samples from ImageNet classes[42] (illustrated in Fig. 3d) processed by AlexNet[43]. Figure 3e demonstrates that the manifold classification capacity increases along the hierarchy for a fully trained network, with a concomitant decrease in manifold dimension and radius. The dimension undergoes a pronounced decrease, from above 80 in the early layers to about 20 in the last feature layer (Fig. 3f). Manifold radius exhibits a uniform decrease from above 1.4 in the input pixel layer to 0.8 in the feature layer (Fig. 3g).

An important question in the theory of DCNNs is to what extent their impressive performance results from the choice of architecture and nonlinearities prior to training[45–47]. We address this issue by processing the same images using an untrained network (with the same architecture but random weights). An untrained network shows very little improvements in classification capacity and manifold geometry; the residual improvement reflects the effect of the architecture. Another useful baseline is the performance on shuffled manifolds. We repeated our analysis of the fully trained network but shuffled the assignment of images into objects. This shuffling destroys any geometrical structure in the manifolds, leaving only residual capacity due to the finite number of samples. The properties of the shuffled manifolds are constant across the hierarchy, similar to those of the true manifolds in the pixel layers, suggesting that for that layer, the manifolds' variability is so large that their properties are similar to random points. In contrast, the last layers of the trained network exhibit substantial improvement in capacity and geometry, reflecting the emergence of robust object representations.

**Capacity increases along hierarchy in point-cloud manifolds.** How does manifold classification capacity depend on the statistics of images representing each object? To answer those questions we consider manifolds with two levels of variability, low variability manifolds consisting of images with the top 10% score ("top 10%", see Methods) and high variability manifolds consisting of all (roughly 1000) images per class ("full class"). Both manifold types exhibit enhanced capacity in the last layers (Fig. 4a). As the absolute value of capacity depends not only on the geometry of the point-cloud manifold but also on the number of samples per manifold (which differ between those two classes of manifolds), we emphasize the improvement in capacity relative to manifolds with shuffled labels; relative manifold capacity improves from random-points level at the pixel layer to an order of magnitude

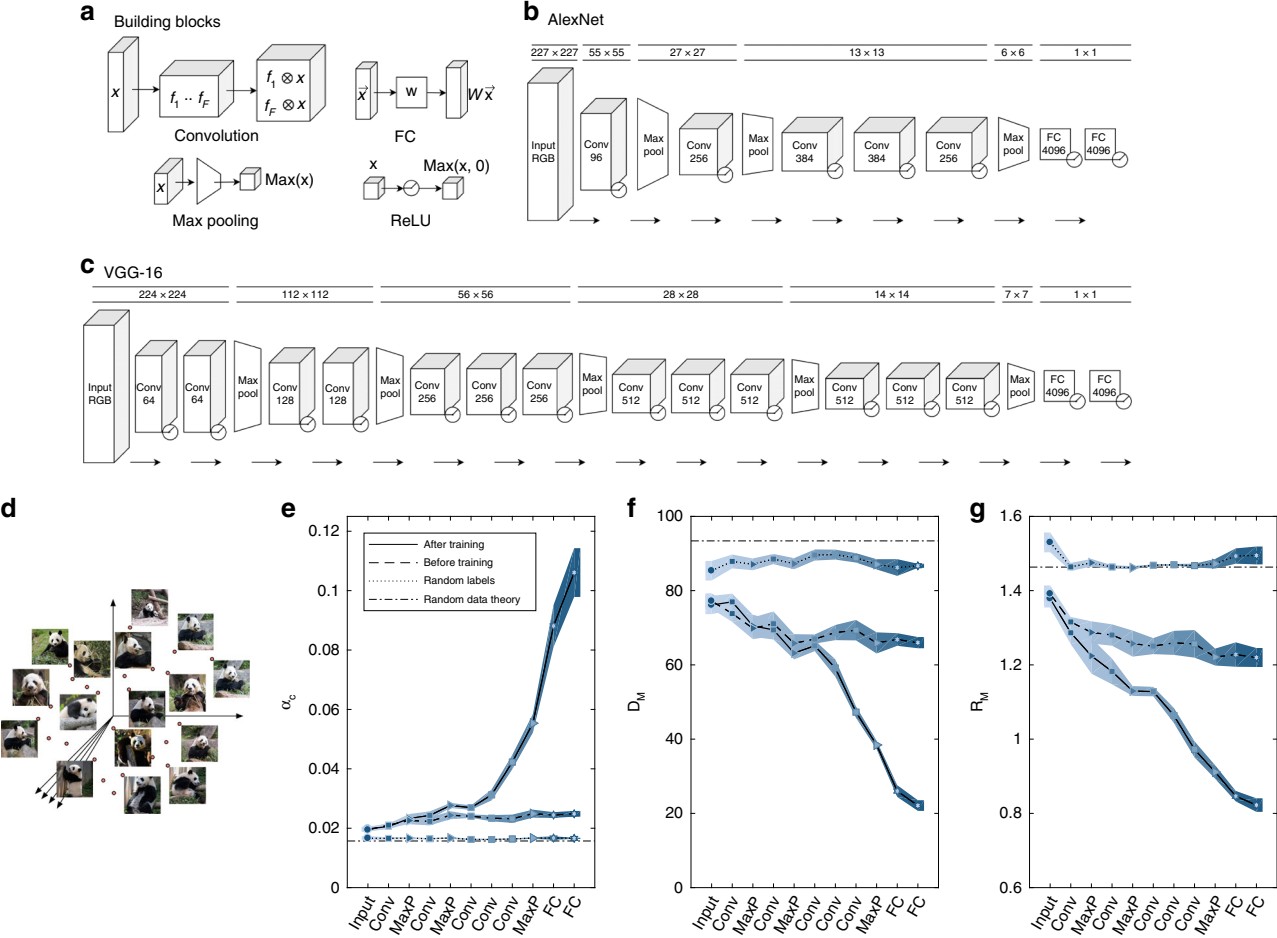

**Fig. 3 Manifolds capacity and geometry changes during learning. a** Illustration of computational building blocks used in AlexNet and VGG-16. Convolution: a set of linear 2-d filters applied on a spacial set of features, each producing a new spacial map. FC: linear fully connected operation (matrix multiplication). Max pooling: a local nonlinearity calculating the maximal activation at small overlapping patches. ReLU: point-wise nonlinearity which sets negative input value to zero. Detailed description of **b** AlexNet and **c** VGG-16 structure, composed of the above building blocks. Dimensions on top describe the spacial dimensions of the layer; the dimension of Conv layers describe the number of convolution filters while the dimension of FC layers is the number of output neurons. ReLU nonlinearity is shown following other operations and is not counted as a layer. **d** Illustration of a point-cloud manifold for the 'giant panda' class, in high-dimensional state space. **e**–**g** Changes in capacity and manifold geometry along the layers of AlexNet point-cloud manifolds (top 10%) for fully trained network (full line), randomly initialized network (dashed line) or randomly shuffled object manifolds (dotted line). Line and markers indicate mean value over five different choices of 50 objects; surrounding shaded areas indicate 95% confidence interval. The value expected by theory for random points (see Methods) is shown as dash-dot line. **e** Changes in classification capacity. **f** Changes in mean manifold dimension. **g** Changes in mean manifold radii. Classification capacity, manifolds dimensions and radii were measured using mean-field theory (see Methods). The x-axis labels provides abbreviation of the layer types (Input—pixel layer, Conv—convolutional layer, MaxP—max-pooling layer, FC—fully connected layer). Marker shape represents layer type (circle—pixel layer, square—convolution layer, right-triangle—max-pooling layer, hexagon—fully connected layer). Features in linear layers (Conv, FC) are extracted after a ReLU nonlinearity.

above it in the last layers. Interestingly, although the high score manifolds exhibit higher absolute capacity than the full manifolds (Supplementary Fig. 1), as the former have fewer points and less variability, the relative improvement is actually larger in the full manifolds.

How does the capacity vary between different DCNNs trained to preform the same object recognition task? Figure 4b shows the corresponding capacity results for a deeper DCNN, VGG-16[44]. Despite difference in the architecture the pattern of improved capacity is quite similar, with most of the increase taking place in the last layers of the networks. Notably, the deeper network exhibits higher capacity in the last layers compared to the shallower network, consistent with the improved performance of VGG-16 in the ImageNet task (this trend continues further with more recent DCNNs, residual networks[48], Supplementary Figs. 1, 2).

**Capacity increases along hierarchy in smooth manifolds**. We now turn to consider manifolds which naturally arise when stimuli has several latent parameters (e.g., translation or distortions) which are smoothly varied. We choose a set of 'template' images (containing different objects, again from the ImageNet data-set[42]) and warp them by multiple affine transformations (see illustration at Fig. 5a, and Methods), resulting in a set of smooth manifolds, each associated with a single template image. Manifolds created by imposing smooth variations of single images are often used in neuroscience experiments[49]. Such manifolds are computationally easy to sample densely, thus allowing us to extrapolate to the case of infinite number of samples, Supplementary Fig. 3). Furthermore, we can independently manipulate the intrinsic manifold dimension (controlled by the number of affine transform parameters) and the manifold extent (controlled by the maximal distortion of the object at the pixel layer, see Methods).

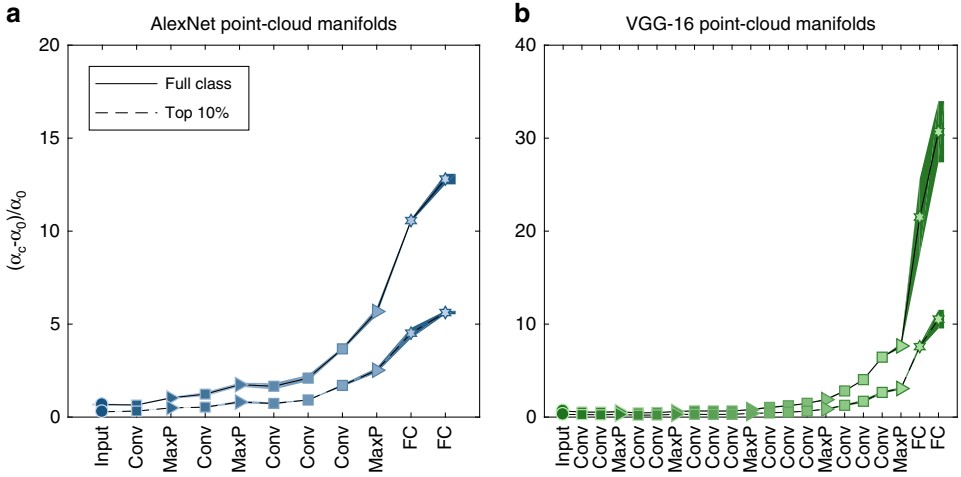

**Fig. 4 Capacity of point-clouds manifolds of ImageNet classes. a, b** Normalized classification capacity for point-cloud manifolds of ImageNet classes (full line: full class manifolds; dashed line: top 10% manifolds) along the layers of AlexNet (**a**) and VGG-16 (**b**). AlexNet top 10% manifolds results already appeared as "after training" results from Fig. 3e. Line and markers indicate mean value over 5 different choices of 50 objects; surrounding shaded areas indicate 95% confidence interval. Capacity is normalized by $\alpha_c = 2/\langle M_\mu \rangle_\mu$, the value expected for unstructured manifolds (see main text; $M_\mu$ denotes the number of samples from object $\mu$). The x-axis labels provides abbreviation of the layer types. Marker shape represents layer type (circle—pixel layer, square—convolution layer, right-triangle—max-pooling layer, hexagon—fully connected layer). Features in linear layers are extracted after a ReLU nonlinearity. Color (blue—AlexNet, green—VGG-16) changes from dark to light along the network.

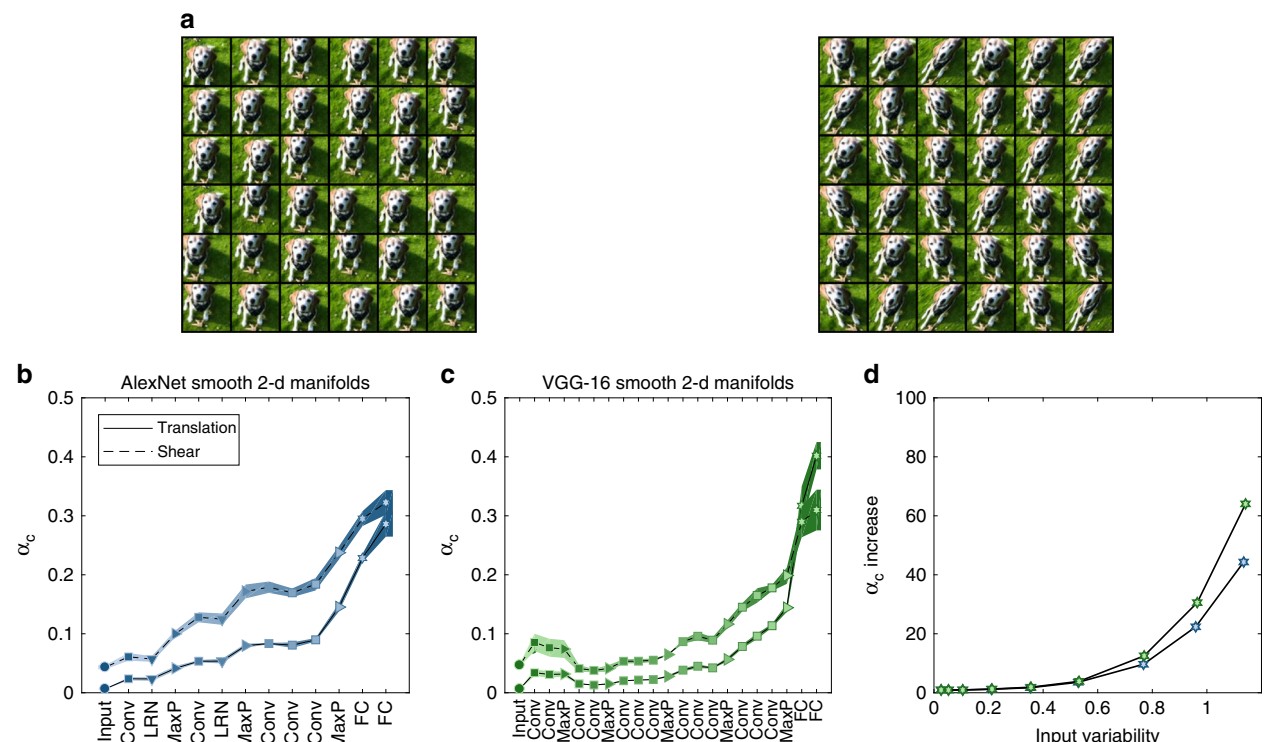

**Fig. 5 Capacity of smooth manifolds from warped ImageNet images. a** Illustration of smooth, densely sampled affine transformed images; 36 samples from a 2-d translation manifold (left) and 2-d shear manifold (right). Each manifold sample is associated with a coordinate specifying the horizontal and vertical translation or shear of a base image, and corresponds to an image where the object is warped using the appropriate affine transformation. **b, c** Classification capacity for 2-d smooth manifolds (full line: translation; dashed line: shear) along the layers of AlexNet (**b**) and VGG-16 (**c**). Line and markers indicate mean value over four different choices of 64 objects; surrounding shaded areas indicate 95% confidence interval. The x-axis labels provides abbreviation of the layer types. Marker shape represents layer type (circle—pixel layer, square—convolution layer, right-triangle—max-pooling layer, hexagon—fully connected layer, down-triangle—local normalization). Features in linear layers are extracted after a ReLU nonlinearity. Color (blue—AlexNet, green—VGG-16) changes from dark to light along the network. **d** Capacity increase from the input (pixel layer) to the output (features layer) of AlexNet (blue markers) and VGG-16 (green markers) for 2-d translation smooth manifolds. The capacity increase is specified as ratio of capacity at the last layer relative to the pixel layer (y-axis), at different levels of stimuli variation measured using Supplementary Eq. (3) at the pixel layer (x-axis).

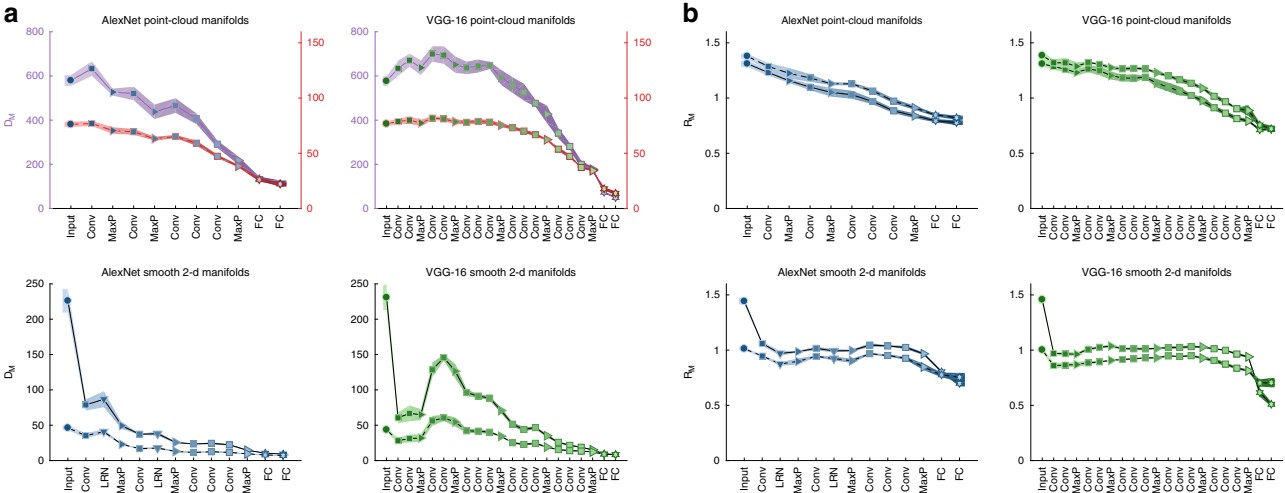

**Fig. 6 Manifold geometry. a** Mean manifold dimension for point-cloud manifolds of AlexNet and VGG-16 (top, full line: full-class manifolds, dashed line: top 10% manifolds) and smooth 2-d manifolds for the same deep networks (bottom, full line: translation manifolds, dashed line: shear manifolds). AlexNet top 10% manifolds results already appeared as "after training" results from Fig. 3f. Values of point-cloud top 10% manifolds are showed against a secondary y-axis (color-coded by the markers edge) to improve visibility. **b** Mean manifold radius for point-cloud manifolds of AlexNet and VGG-16 (top, full line: full-class manifolds, dashed line: top 10% manifolds) and smooth 2-d manifolds for the same deep networks (bottom, full line: translation manifolds, dashed line: shear manifolds). AlexNet top 10% manifolds results already appeared as "after training" results from Fig. 3g. Line and markers indicate mean value over different choices of objects; surrounding shaded areas indicate 95% confidence interval. The x-axis labels provides abbreviation of the layer types. Marker shape represents layer type (circle—pixel layer, square—convolution layer, right-triangle—max-pooling layer, hexagon—fully connected layer, down-triangle—local normalization layer). Features in linear layers are extracted after a ReLU nonlinearity. Color (blue—AlexNet, green—VGG-16) changes from dark to light along the network.

A substantial improvement of classification capacity of smooth manifolds is observed along the networks with a pronounced increase in the last layers (Fig. 5b, c), similar to the behavior observed for point-cloud manifolds. The relative increase in capacity from the first (pixel) layer to the last layer is shown in Fig. 5d for manifolds with different variability levels. Notably this increase is growing faster than manifold variability itself, reaching an increase of almost two orders of magnitude for the highest variability considered here. This holds for both 1-d and 2-d smooth manifolds and in all deep networks considered, including ResNet-50 (Supplementary Fig. 4).

**Network layers reduce dimension, radius of object manifolds**. Changes in the measured classification capacity can be traced back to changes in manifold geometry along the network hierarchy, namely manifold radii and dimensions, which can be estimated from data (see Methods, Eqs. (3) and (4), and Supplementary Methods 2.1). Mean manifold dimension and radius along DCNNs hierarchies are shown in Fig. 6a, b, respectively. The results exhibit a surprisingly consistent pattern of changes in the geometry of manifolds between different network architectures, along with interesting differences between the behavior of point-cloud and smooth manifolds. Figure 6a (and Supplementary Fig. 5 for ResNet-50 results) suggests that decreased dimension along the deep hierarchies is the main source of the observed increase in capacity from Figs. 4 and 5. Both point-cloud and smooth manifolds exhibit non-monotonic behavior of dimension, with increased dimension in intermediate layers; this increase of dimensionality is also be observed in other measures such as participation ratio (Supplementary Fig. 5). A notable result is the very pronounced decrease in dimensions after the pixel layer of smooth translation manifolds (Fig. 6a, bottom), consistent with the expected ability of this convolution layer to average out substantial variability in images due to translation. On the other hand, manifold radii undergo modest decrease along the deep hierarchy and across all manifolds (Fig. 6b, and

Supplementary Fig. 6 for ResNet-50). The larger role of dimensions, rather than radii, in contributing to the increase in capacity is demonstrated by comparing the observed capacity to that expected for manifolds with the observed dimensions but radii fixed at their value at the pixel layer, or the other way around (Supplementary Fig. 7). Interestingly, the decrease in radius is roughly linear in point-cloud manifolds while for smooth manifolds we find a substantial decrease in the first layer and the final (fully connected) layers, but not in intermediate layers. Those differences may reflect the fact that the high variability of point-cloud manifolds needs to be reduced incrementally from layer to layer (both in terms of radius and dimension), utilizing the increased complexity of downstream features, while the variability created by local affine transformations is handled mostly by the local processing of the first convolutional layer (consistent with ref. [35] reporting invariance to such transformations in the receptive field of early layers). The layer-wise compression of affine manifold plateaus in the subsequent convolutional layers, as the manifolds are already small enough. As signals propagate beyond the convolutional layers, the fully connected layers add further reduction in size in both manifold types.

This geometric description allows us to further shed light on the structure of the smooth manifolds used here. For radius up to 1, the dimension of the manifolds with intrinsic 2-d variations (e.g., created by vertical and horizontal translation) is just the sum of the dimensions of the two corresponding 1-d manifolds with the same maximal object displacement (Supplementary Fig. 8a); only for larger radii, dimensions for 2-d manifolds are super-additive. On the other hand, for all levels of stimulus variability the radius of 2-d manifolds is about the same as the value of the corresponding 1-d manifolds (Supplementary Fig. 8b). This highlights the non-linear structure of those larger manifolds, where the effect of changing multiple manifold coordinates is no longer predicted from the effect of changing each coordinate separately.

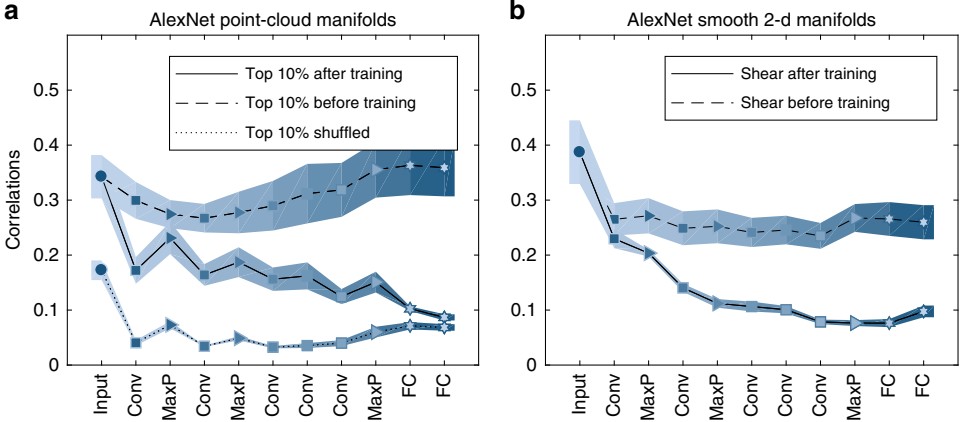

**Fig. 7 Correlations between manifolds.** Changes of mean between-manifold correlations along the layers of AlexNet. **a** Center correlations for top 10% point-cloud manifolds in fully trained network (full line), randomly initialized network (dashed line) or randomly shuffled object manifolds (dotted line). **b** Center correlations for smooth 2-d shear manifolds in fully trained network (full line) or randomly initialized network (dashed line). Line and markers indicate mean value over different choices of objects; surrounding shaded areas indicate 95% confidence interval. The *x*-axis labels provides abbreviation of the layer types. Marker shape represents layer type (circle—pixel layer, square—convolution layer, right-triangle—max-pooling layer, hexagon—fully connected layer). Features in linear layers are extracted after a ReLU nonlinearity. Color changes from dark to light along the network. Center correlations are $\rho_{CC} = <|\vec{x}'^\mu \cdot \vec{x}'^\nu|/||\vec{x}'^\mu|| \cdot ||\vec{x}'^\nu||>_{\mu \neq \nu}$, where $\vec{x}'^\mu$ is the center of object $\mu$ (Supplementary Eq. (1)).

**Network layers reduce correlations between object centers.** Manifold geometry considered above characterizes the variability in object manifolds' shape but not the possible relations between them. Here we focus on the correlations between the centers of different manifolds (hereafter: center correlations), which may create clusters of manifolds in the representation state space. Though clustering may be beneficial for specific target classifications, our theory predicts that the overall effect of such manifold clustering on random binary classification is detrimental. Hence, these correlations reduce classification capacity (Supplementary Note 3.1). Thus, the amount of center correlations at each layer of a deep network is a computationally-relevant feature of the underlying manifold representation.

Importantly, for both point-cloud and smooth manifolds we find that in an AlexNet network trained for object classification, center correlations decrease along the deep hierarchy (full lines in Fig. 7a, b; additional VGG-16, ResNet-50 results in Supplementary Fig. 9). This decrease is interpreted as incremental improvement of the neural code for objects, and supports the improved capacity (Figs. 4–5). In contrast, center correlations at the same network architectures but prior to training (dashed lines in Fig. 7a, b) do not decrease (except for the affine manifolds in the first convolutional layer, Fig. 7b). Thus this decorrelation of manifold centers is a result of the network training. Interestingly, the center correlations of shuffled manifolds exhibit lower levels of correlations, which remain constant across layers after an initial decrease at the first convolutional layer.

Another source of inter-manifold correlations are correlations between the axes of variation of different manifolds; those also decrease along the network hierarchies (Supplementary Fig. 9) but their effect on classification capacity is small (as verified by using surrogate data, Supplementary Fig. 10).

**Effect of network building blocks on manifolds' geometry.** To better understand the enhanced capacity exhibited by DCNNs we study the roles of the different network building blocks. Based on our theory, any operation applied to a neural representation may change capacity by either changing the manifolds' dimensions, radii, or the inter-manifold correlations (where a reduction of these measures is expected to increase capacity).

Figure 8a, b shows the effect of single operations used in AlexNet and VGG-16. We find that the ReLU nonlinearity usually reduces center correlations and manifolds' radii, but increases manifolds' dimensions (Fig. 8a). This is expected as the nonlinearity tends to generate a sparse, higher dimensional, representations[50,51]. In contrast, pooling decreases manifolds' radii and dimensions but usually increase correlations (Fig. 8b), presumably due to the underlying spatial averaging. Such clear behavior is not evident when considering convolutional or fully connected operations in isolation (Supplementary Fig. 11).

In contrast to single operations, we find that the networks' computational building blocks perform consistent transformation on manifold properties (Fig. 8c, d). The initial building blocks consist of sequences of convolution, ReLU operation followed by pooling, which consistently act to decrease correlations and tend to decrease both manifolds' radii and dimensions (Fig. 8c). On the other hand, the final building block, a fully connected operation followed by ReLU, decreases manifolds' radii and dimensions, but may increase correlations (Fig. 8d), similarly to the max-pooling operation (Fig. 8b). Furthermore, as composite building blocks show more consistent behavior than individual operations, we understand why DCNNs with randomly initialized weights do not improve manifold properties. Only by appropriately trained weights, the combination of operations with often opposing effects yields a net improvement in manifold properties.

**Comparison of theory with numerically measured capacity.** The results presented so far were obtained using algorithms derived from a mean-field theory which is exact in the limit of large number of neurons and manifolds and additional simplifying statistical assumptions (Supplementary Note 3.1). To test the agreement between the theory and the capacity of finite-sized networks with realistic data, we have numerically computed capacity at each layer of the network, using recently developed efficient algorithms for manifold linear classification[41] (see Methods). Comparing the numerically measured values to theory shows good agreement for both point-cloud manifolds (Fig. 9a) and smooth manifolds (Fig. 9b, Supplementary Fig. 12). This agreement is a remarkable validation of the applicability of mean-field theory to representations of realistic datasets generated by complex networks.

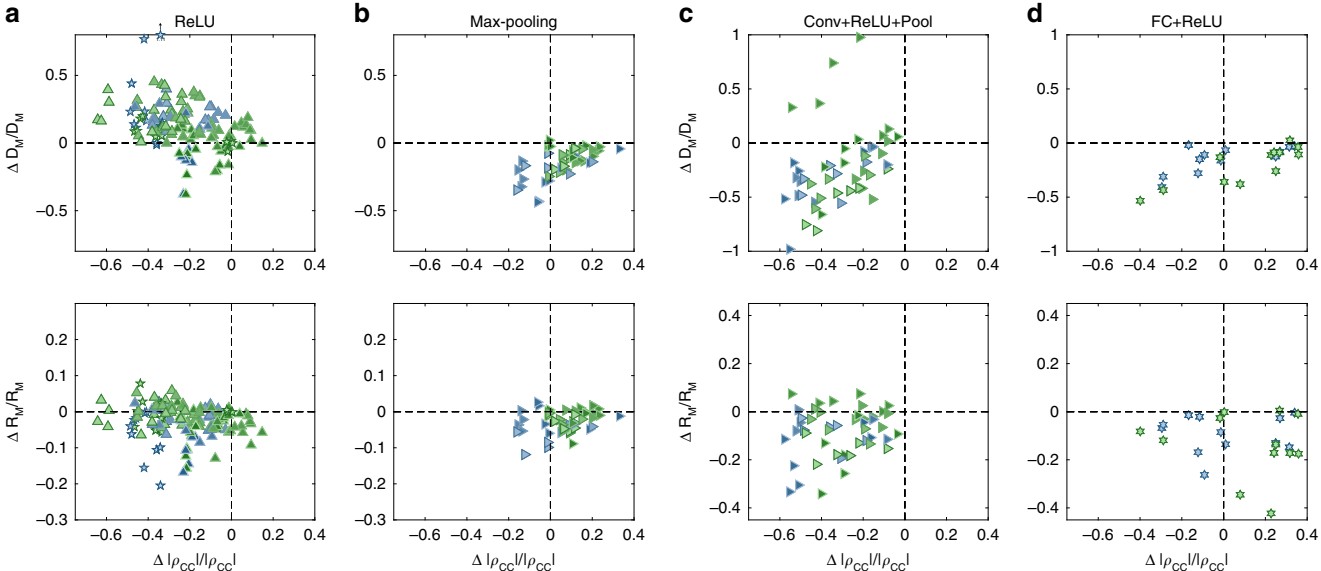

**Fig. 8 Manifold property changes by network building blocks.** Changes in the relative manifold properties between the input and the output of different network building blocks, shown as change in dimension vs change in center correlations (top) and change in radius vs change in center correlations (bottom). Each panel pools results from a specific building block in AlexNet (blue markers) and VGG-16 (green markers) for both point-cloud manifolds (full class, top 10%) and smooth manifolds (1-d and 2-d, translation and shear). Marker shape represents layer type (square—convolution layer, right-triangle—max-pooling layer, hexagon—fully connected layer). For layer sequences marker shape represents the last layer in the sequence. For isolated ReLU marker shape represent previous layer type; pentagon—ReLU after fully connected layer, up-triangle—ReLU after convolution layer. Color changes from dark to light along the network. **a** Changes in manifold properties for isolated ReLU operations. **b** Changes in manifold properties for isolated Max-pooling operations. **c** Changes in manifold properties for a common sequence of operations: one or more repetitions of convolution, ReLU operations, with or without intermediate normalization operation, ending with a max-pooling operation. **d** Changes in manifold properties for a common sequence of operations: fully connected, ReLU operations. The data analyzed here correspond to the first set of objects used in the analysis of the mean and confidence interval presented in Figs. 6 and 7, and include additional intermediate values not presented in those plots, notably the inputs of ReLU operations.

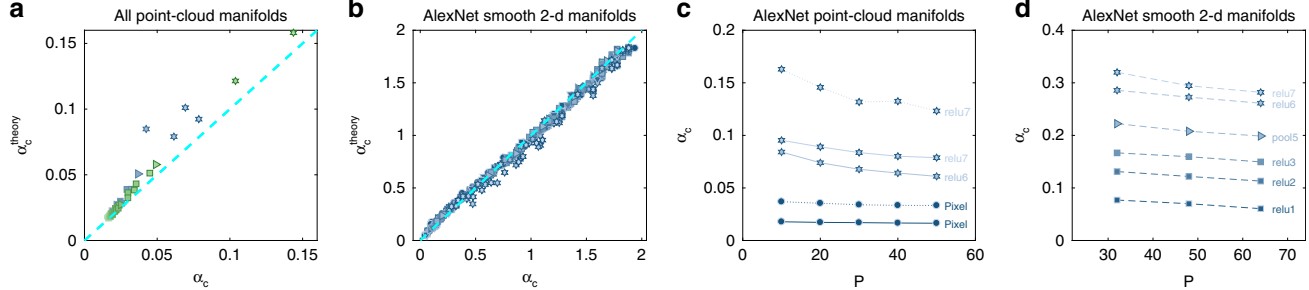

**Fig. 9 Theoretical predictions. a** Comparison of numerically measured capacity ($x$-axis) with the theoretical prediction ($y$-axis) for AlexNet, VGG-16 at different layers along the hierarchy (top 10% point-cloud manifolds). **b** Comparison of numerically measured capacity ($x$-axis) with the theoretical prediction ($y$-axis) for AlexNet at different layers along the hierarchy and different levels of manifold variability (smooth 2-d manifolds). **c** Numerically measured capacity ($y$-axis) at different number of objects ($x$-axis) for point-cloud manifolds at different layers (dashed line: top 10% manifolds; dotted line: top 5% manifolds). **d** Numerically measured capacity ($y$-axis) at different number of objects ($x$-axis) for smooth 2-d shear manifolds. Marker shape represents layer type (circle—pixel layer, square—convolution layer, right-triangle—max-pooling layer, hexagon—fully connected layer, down-triangle—local normalization layer). Color (blue—AlexNet, green—VGG-16) changes from dark to light along the network.

A fundamental prediction of the theory is that the maximal number of classifiable manifolds $P_c$ is extensive, namely grows in proportion to the number of neurons in the representation $N$, hence their ratio $\alpha_c$ is unchanged. We validated this prediction in realistic manifolds, by measuring numerically the capacity upon changing both the number of neurons used for classification and the number of data manifolds. Capacity for both point-cloud manifolds (Fig. 9c) and smooth manifolds (Fig. 9d) exhibits only a modest dependence on the number of objects on which it is measured, and seems to saturate at a finite value of $P \approx 50$

(additional results for 1-d and 2-d smooth manifolds with different variability levels are provided in Supplementary Fig. 13).

Note that the mean-field prediction of extensivity of classification holds for manifold ensembles whose individual geometric measures such as manifold dimension and radius do not scale with the representation size but retain a finite limit when $N$ grows. Indeed, we found that the radii and dimensions of our data manifolds also show little dependence on the number of neurons for values of $N$ larger than several hundred (Supplementary Fig. 14), consistent with the exhibited extensive capacity. The

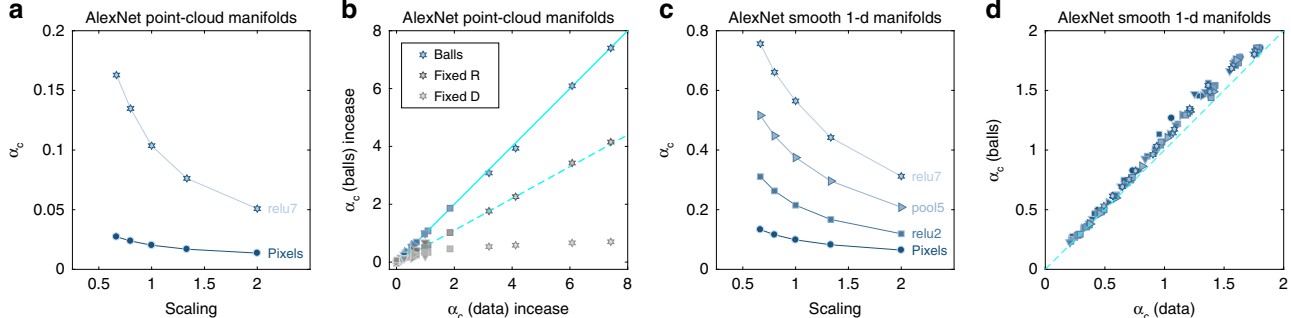

**Fig. 10 Manifold structure perturbations effect on capacity.** **a** Classification capacity following manifold scaling (x-axis indicate scaling factor, with 1 corresponds to no scaling) for AlexNet at different layers along the hierarchy (top 10% point-cloud manifolds). **b** Comparison of classification capacity (x-axis) with the prediction from balls with the same manifold properties (y-axis) for AlexNet at different layers along the hierarchy (point-clouds of top 10% and full class manifolds). As capacity of those manifolds spans two orders of magnitude it is normalized by capacity at the pixel layer. The full cyan line indicate $y = x$ while the dashed cyan line indicate $y = 0.55x$. **c** Classification capacity following manifold scaling (x-axis indicate scaling factor, with 1 corresponds to no scaling) for AlexNet at different layers along the hierarchy (smooth 1-d translation manifolds). **d** Comparison of numerically measured capacity (x-axis) with numerically measured capacity of balls with the same manifold properties (y-axis) for AlexNet at different layers along the hierarchy and different levels of manifold variability (smooth 1-d translation manifolds). The dashed cyan line indicate $y = x$. Marker shape represents layer type (circle—pixel layer, square—convolution layer, right-triangle—max-pooling layer, hexagon—fully connected layer, down-triangle—local normalization layer). Color changes from dark to light along the network.

saturation of $\alpha_c$ and manifold geometry with respect to $N$ implies that they can be estimated on a subsampled set of neurons (or subset of random projections of the representation) at each layer (see Methods), a fact that we utilized in calculating our results of Figs. 3–8 and has important practical implications for the applicability of these measures to large networks.

**Effect of manifold perturbations on capacity**. So far, we have shown that deep networks enhance manifold classification capacity, while reducing their dimensions, radii and correlations. But can we show that manifold dimension and radius, rather than other salient shape features, are indeed causally related to the increase in classification capacity? Here we utilize the full accessibility of the representations in artificial neural networks to address these questions, by manipulating the geometry of the manifold representations. First, we show that increasing the size of the manifolds without changing other geometric features is sufficient to decrease capacity. We multiply all vectors belonging to a manifold relative to the manifold center by a global scaling factor. Figure 10a, c displays the decrease of capacity with the scaling factor for both point-cloud and smooth manifolds at several representative layers in AlexNet.

To show that the changes in manifold dimensions and radii are sufficient to explain the observed changes in capacity, we recalculated the capacity by replacing the manifolds at each layer with balls with the same radius and dimension as the $R_M$ and $D_M$, respectively. Figure 10b, d shows a remarkable agreement between the balls and the original manifolds, proving that $R_M$ and $D_M$ are the dominant geometric measures underlying the improved capacity. To quantitatively estimate the relative contributions of reductions in dimensions and radii to the enhanced capacity, we compared the actual capacity in different layers with that of balls with the same dimension as the manifold dimension in the respective layer but with radius fixed at its pixel layer value (as in Supplementary Fig. 7). Figure 10b shows a substantial improvement in capacity ranging between 90% of the full capacity in early layers and 55% in last layers. In contrast, the capacity of balls with the same radius as the manifolds but with dimension fixed to their pixel layer value, exhibits only a small improvement, supporting our conclusion that the dominant factor in improved manifold separability is the reduction in their dimensions.

Finally, to demonstrate the causal role of center correlations on capacity, we have manipulated the manifold centers by randomizing them without changing the geometry of the manifolds, and compared the resultant capacity (Supplementary Fig. 10b). As anticipated, center randomization improves capacity, especially in the first layers where the actual capacity exhibit high center correlations.

## Discussion

The goal of our study was to delineate the contributions of computational, geometric, and correlation-based measures to the untangling of manifolds in deep networks, using a new theoretical framework. To do this, we introduce classification capacity as a measure of the amount of decodable categorical information per neuron. Combining tools from statistical physics and insights from high-dimensional geometry, we develop a mean-field estimate of capacity and relate it to geometric measures of object manifolds as well as their correlation structure. Using these measures, we analyze how manifolds are reformatted as the signals propagate through the deep networks to yield an improved invariant object recognition at the last stages.

We find that the classification capacity of the object manifolds increases across the layers of trained DCNNs. At the pixel layer, the extent of intra-manifold variability is so large that the resultant manifold properties are almost indistinguishable from random points. Subsequent processing by the trained networks results in a significant increases in capacity along with an overall reduction in the mean manifold dimensions and radii. In contrast, networks with the same architectures but random weights do exhibit only slight improvement in capacity and manifold geometry, indicating that training rather than mere architecture is responsible for the successful reformatting of the manifolds.

For both point-cloud and smooth manifolds across multiple DCNNs architectures (AlexNet, VGG-16, and ResNet-50), we find improved manifold classification capacity (Figs. 4 and 5, Supplementary Figs. 1, 2, 4) associated with decreased manifold dimension and radius across the layers (Fig. 6, Supplementary Figs. 5, 6). Our findings suggest that different network hierarchies employ similar strategies to process stimulus variability, with image warping handled by the initial convolution and last layers, and intermediate layers needed to gradually reduce the dimension and radius of point-cloud manifolds. We find that lowering the

dimensionality of each object manifold is one of the primary reasons for the improved separation capabilities of the trained networks.

The networks also effectively reduce the inter-manifold correlations, and some of the improved capacity exhibited by the DCNNs results from decreased manifold center correlations across the layers (Fig. 7, Supplementary Fig. 9). As with the manifold geometrical measures, the improved decorrelation is specific to networks with trained weights; for random weights, center correlations remain high across all layers. Other studies of object representations in DCNNs and in the visual system[3,7,10] focused on the comparison between the correlational structures of representations in different systems (e.g., different networks, or animals vs humans). Here we find that the deep networks exhibit an overall decrease of correlations between object manifolds and demonstrate its computational significance. Decorrelation between neuronal responses to natural stimuli and the associated redundancy reduction has been one of earliest principles proposed to explain principles of neural coding in early stages of sensory processing[50,52–55]. Interestingly, here we find that decorrelation between object representations is an important computational principle in higher processing stages.

In this work we have not addressed the question of extracting physical variables of stimuli, such as pose or articulation. In principle, reformatting of object manifolds might also involve alignment of their axes of variation, so that information about physical variables can be easily readout by projection on subspace orthogonal to directions which contain object identity information[56]. Alternatively, separate channels may be specialized for such tasks. Interestingly, in artificial networks, the axes–axes alignment across manifolds is reduced after the first layers (Supplementary Fig. 9), consistent with their training to perform only object recognition tasks. This is qualitatively consistent with the study of information processing in deep networks[57] which proposes a systematic decrease along the network hierarchy in information about the stimulus accompanied by increased representation of task related variables. It would be interesting to examine systematically if high level neural representations in the brain such as IT cortex show similar patterns or channel both type of information in separate dimensions[11,58].

Our analysis of the effect of common computational building blocks in DCNNs (such as convolution, ReLU nonlinearity, pooling and fully connected layers) shows that single stages do not explain the overall improvement in manifold structure. Some individual stages transform manifold geometry differently dependent on their position in the network (e.g., convolution, Supplementary Fig. 11). Other stages exhibit trade-offs between different manifold features; for instance, the ReLU nonlinearity tends to reduce radius and correlations but increase the dimensionality. In contrast, composite building blocks, comprising a sequence of spatial integration, local nonlinearities and non-local pooling yield a consistent reduction in manifold radius and dimension in addition to reduced correlations across the different manifold types and network architectures.

We find very similar behavior in manifolds propagated through another class of deep networks, residual networks, that are not only much deeper but also incorporate a special set of skip connections between consecutive modules. Residual networks with different number of layers exhibit consistent behavior under our analysis (Supplementary Fig. 2). Focusing on ResNet-50 (Supplementary Figs. 1, 4, 5, 6, 8, 9), we find quite similar behavior to the networks in Figs. 3–7. Furthermore, on this architecture, each skip module exhibits consistent reductions in manifold dimensions, radii and correlations (Supplementary Fig. 11c), similar to the changes in the other network architectures (Fig. 8).

Consistent across all the networks we studied, the increase in capacity is modest for most of the initial layers and improves considerably in the last stages (typically after the last convolution building block). This trend is even more pronounced for residual networks (Supplementary Figs. 1–4). This does not imply that previous stages are not important. Instead, it reflects the fact that capacity intimately depends on the incremental improvement of a number of factors including geometry and correlation structure.

Given the ubiquity of the changes in the manifold representations found here, we predict that similar patterns will be observed for sensory hierarchies in the brain. One issue of concern is trial-to-trial variability in neuronal responses. Our analysis assumes deterministic neural responses with sharp manifold boundaries, but it can be extended to the case of stochastic representations where manifolds are not perfectly separable. Alternatively, one can interpret the trial averaged neural responses as representing the spatial averaging of the responses of a group of stochastic neurons, with similar signal properties but weak noise correlations. To properly assess the properties of perceptual manifolds in the brain, responses of moderately large subsampled populations of neurons to numerous objects with multiple sources of physical variability is required. Such datasets are becoming technically feasible with advanced calcium imaging[27]. Recent work has also enabled quantitative comparisons to DCNNs from electrophysiological recordings from V4 and IT cortex in macaques[59].

One extension of our framework would relate capacity to generalization, the ability to correctly classify test points drawn from the manifolds but not part of the training set. While not addressed here, we expect that it will depend on similar geometric manifold measures, namely stages with reduced $R_M$ and $D_M$ will exhibit better generalization ability. Those geometric manifold measures can be related to optimal separating hyperplanes which are known to provide improved generalization performance in support vector machines.

The statistical measures introduced in this work (capacity, geometric manifold properties, and center correlations) can also be used to guide the design and training of future deep networks. By extending concepts of efficient coding and object representations to higher stages of sensory processing, our theory can help elucidate some of the fundamental principles that underlie hierarchical sensory processing in the brain and in deep artificial neural networks.

## Methods

**Summary of manifold classification capacity and anchor points.** Following the theory introduced in ref. [36], manifolds are described by $D + 1$ coordinates, one which defines the location of the manifold center and the others the axes of the manifold variability. The set of points that define the manifold within its subspace of variability is formally designated as $\mathcal{S}$ which can represent a collection of finite number of data points or a smooth manifold (e.g., a sphere or a curve). An ensemble of $P$ manifolds is defined by assuming the center locations and the axes' orientations are random (focusing first on the case where all manifolds have the same shape). Near capacity the separating weight vector can be decomposed into at most $P$ representative vectors, one from each manifold, such that $\mathbf{w} = \sum_{\mu=1}^{P} \lambda_\mu y^\mu \bar{\mathbf{x}}^\mu$, where $\lambda_\mu \geq 0$ and $\bar{\mathbf{x}}^\mu \in \mathrm{conv}(M^\mu)$ is a representative vector in the convex hull of $M^\mu$, the $\mu$-th manifold. These vectors play a key role in the theory, as they comprehensively determine the separating plane. We denote these representative points from each manifold as the manifold anchor points.

Classification capacity is defined as $\alpha_c = P_c/N$ where $P_c$ is the maximum number of manifolds that can be linearly separated using random binary labels. In mean-field theory, capacity is described in terms of a self-consistent equations involving a single manifold embedded in an ensemble of many others. These equations takes the form of

$$\alpha_c^{-1} = \langle F(\vec{T}) \rangle_{\vec{T}} \tag{1}$$

$$F(\vec{T}) = \min_{\vec{V}} \left\{ \left\| \vec{V} - \vec{T} \right\|^2 \middle| \min_{\vec{S}} \left\{ \vec{V} \cdot \vec{S} \middle| \vec{S} \in \mathcal{S} \right\} \geq 0 \right\} \tag{2}$$

where $\langle \dots \rangle_{\overrightarrow{T}}$ is an average over random $D+1$-dimensional vector $\overrightarrow{T}$ whose components are i.i.d. normally distributed $T_i \sim \mathcal{N}(0,1)$. The components of the vector $\overrightarrow{V}$ represent the signed fields induced by the separating vector $\mathbf{w}$ (near capacity) on the axes of a single manifold, e.g., $M^\mu$. The inequality constraints on $\overrightarrow{V}$ in Eq. (2) ensures that the projections of all the points on $\mathbf{w}$ are positive, so that all the points on the manifold are correctly classified. The projected weight vector on a manifold, $\overrightarrow{V}$ is a sum of two contributions, one comes from the manifold's own anchor point $\lambda_\mu y^\mu \tilde{\mathbf{x}}^\mu$ and another from the random projections of all other anchor points on the subspace of $M^\mu$. In the limit of large $N$ and $P$, these projections are normally distributed, and are denoted by a $D+1$ Gaussian vector $\overrightarrow{T}$. In order to allow for maximal number of manifolds to be separated $\mathbf{w}$ has to be such that there is maximal agreement between $\overrightarrow{V}$ and the contributions from the other manifolds $\overrightarrow{T}$, hence the minimization with respect to $\left\| \overrightarrow{V} - \overrightarrow{T} \right\|^2 = \left\| \lambda_\mu y^\mu \tilde{\mathbf{x}}^\mu \right\|^2$ in Eq. (2). Finally, due to a fixed square norm of the weight vector, chosen to be $N$, the total contributions from all manifolds, which are on average $P\langle F(\overrightarrow{T})\rangle_{\overrightarrow{T}}$ sums to $N$, yielding Eq. (1) (details can be found in ref. [36]). Note that in the mean-field theory, the representation size $N$ does not appear.

**Geometric properties of manifolds.** For a given manifold, its anchor point $\tilde{\mathbf{x}}^\mu$ depends on the other manifolds in the ensemble. In the mean-field theory, summarized above, this dependence is captured statistically, by the dependence of the anchor point projection on the manifold subspaces, denoted by $\tilde{S}$ on the random vector $\overrightarrow{T}$, representing the random contributions to the separating plane from other manifolds. Thus, the Gaussian statistics of $\overrightarrow{T}$ induces a statistical measure on the manifold's anchor points (projected on the manifolds subspcaces) $\tilde{S}(\overrightarrow{T})$. Since the anchor points determine the separating hyperplane, their statistics, and in particular the induced effective radius and dimension, plays an important role in the classification capacity.

The effective radius and dimension are specified in terms of $\delta\tilde{S} = (\tilde{S} - S_0)/\|S_0\|$, the projection of the anchor point $\tilde{\mathbf{x}}$ onto the $D+1$-dimensional subspace of each manifold, $\tilde{S}$, relative to the manifold center, $S_0$, capturing the statistics of the variation of the points in the $D$-dimensional subspace of manifold variability. Here, the variation of these points is normalized by the manifold center norm; as the centers are random this is equivalent, up to a constant, to normalizing by the average distance between the manifold centers. Then the manifold radius is the total variance of the normalized anchor points,

$$R_M^2 = \left\langle \left\| \delta\tilde{S}(\overrightarrow{T}) \right\|^2 \right\rangle_{\overrightarrow{T}} \qquad (3)$$

The effective dimension quantifies the spread of the anchor points along the different manifold axes and is defined by the angular spread between $\overrightarrow{\delta T} = \overrightarrow{T} - T_0$ (where $T_0$ is $\overrightarrow{T}$ projected on the center $S_0$) and the corresponding anchor point $\delta\tilde{S}(\overrightarrow{T})$ in the manifold subspace:

$$D_M = \left\langle \left( \overrightarrow{\delta T} \cdot \hat{\delta S}(\overrightarrow{T}) \right)^2 \right\rangle_{\overrightarrow{T}} \qquad (4)$$

where $\hat{\delta S}$ is a unit vector in the direction of $\delta\tilde{S}$. In the case where the manifold has an isotropic shape, $D_M = \left\langle \left\| \delta\overrightarrow{T} \right\|^2 \right\rangle = D$ (for detailed derivation see Section 4-D of ref. [36]).

Importantly, the theory provides a precise connection between of manifold capacity and the effective manifold dimensionality and radius, which can be concisely summarized as:

$$\alpha_c \approx \alpha_{Ball}(R_M, D_M) \qquad (5)$$

where $\alpha_{Ball}(R, D)$ is the expression for the capacity of $L_2$ balls with radius $R$ and dimension $D$ (see ref. [60] and Supplementary Eq. (4)). This relation holds for $D_M \gg 1$ and is an excellent approximation for all cases considered here. For a general manifold we interpret the radius as maximal variation per dimension (in units of the manifold's center norm) while the dimension as the number of effective axes of variation, with the manifold's total extent given by $R_M\sqrt{D_M}$.

**Capacity of manifolds with low-rank centers correlation structure.** A theoretical analysis of classification capacity for manifolds with correlated centers is possible using the same tools as in ref. [36], and is provided in Supplementary Note 3.1. Denoting $x^\mu$ the center of mass of manifold $M^\mu$, assuming the $P \times P$ dimensional correlation matrix between manifold centers $C_{\mu\nu} = \langle x^\mu x^\nu \rangle$ satisfies a low-rank off-diagonal structure

$$C = \Lambda + C_K \qquad (6)$$

where $\Lambda$ is diagonal and $C_K$ is of rank $K$, i.e., can be written as $C_K = \sum_1^K c_k \overrightarrow{u}_k \overrightarrow{u}_k^T$. In this case $\{\overrightarrow{u}_k\}_{k=1}^K$ are 'common components, shared across all manifolds', while

$\Lambda_{\mu\mu}$ describe the $\mu$-th manifold's center norm in their null-space. The theory then predicts that for $K \ll P$, the capacity depends on the structure of the manifolds projected to the null-space of the common components (see Supplementary Note 3.1). Thus calculation of capacity in the presence of center correlations requires knowledge of their common components.

**Recovering low-rank centers correlations structure.** In order to take into account the correlations between centers of manifolds that may exist in realistic data, before computing the effective radius and dimension we first recover the common components of the centers by finding an orthonormal set $U \in \mathbb{R}^{N \times K}$ such that the centers projected to its null-space have approximately diagonal correlation structure. Then the entire manifolds are projected into the null-space of the common components. As the residual manifolds have uncorrelated centers, classification capacity is predicted from the theory for uncorrelated manifolds (Eq. (1)).

The validity of this prediction is demonstrated numerically for smooth manifolds in Supplementary Fig. 12. Furthermore, the manifolds geometric properties $R_M$, $D_M$ from Eqs. (3) and (4) are calculated from the residual manifolds using the procedure from ref. [36]. Those are expected to approximate capacity using Eq. (5) when the dimension is substantial; the validity of this approximation for smooth manifolds is demonstrated numerically in Supplementary Fig. 15. The full procedure is described in Supplementary Methods 2.1.

**Inhomogeneous ensemble of manifolds.** The object manifolds considered above may each have a unique shape and size. For a mixture of heterogeneous manifolds[36], classification capacity for the ensemble of object manifolds is given by averaging the inverse of the object manifold capacity estimated from each manifold separately: $\alpha^{-1} = \langle \alpha_\mu^{-1} \rangle_\mu$. Reported capacity value $\alpha_c$ are calculated by evaluating the mean-field estimate from individual manifolds and averaging their inverse over the entire set of $P$ manifolds. Similarly, the displayed radius and dimensions are averages over the manifolds (using a regular averaging). An example of distribution of geometric metrics over the different manifolds is shown in Supplementary Figs. 5 and 6.

**Manifold properties for random manifolds.** A theoretical analysis of classification capacity and geometric properties for a manifold composed of $M$ random points provides a useful baseline for comparison when analysing manifold properties for real-world data. As derived in Supplementary Note 3.2, for this case we expect dimension to scale linearly with the number of random samples (per manifold) $D_M = \frac{\pi}{2(\pi-1)} M$ while the radius is expected to be independent of it $R_M = \sqrt{\pi-1}$. Finally, capacity in this case is expected to be $\alpha_c = \frac{2}{M}$, as predicted by other methods[37,39].

**Measuring capacity numerically from samples.** Classification capacity can be measured numerically by directly performing linear classification of the manifolds. Consider a population of $N$ neurons which represents $P$ objects through their population responses to samples of those objects. Assuming the objects are linearly separable using the entire population of $N$ neurons, we seek the typical sub-population size $n$ where those $P$ objects are no longer separable. For a given sub-population size $n$ we first project the $N$ dimensional response to the lower dimension $n$ using random projections; using sub-sampling rather than random projections provide very similar results but breaks down for very sparse population responses (Supplementary Fig. 16). Second, we estimate the fraction of linearly separable dichotomies by randomly sampling binary labels for the object manifolds and checking if the sub-population data is linearly separable using those labels. Testing for linearly separability of manifold can be done using regular optimization procedures (i.e., using quadratic optimization), or using efficient algorithms developed specifically for the task of manifold classification[41]. As $n$ increase the fraction of separable dichotomies goes from 0 to 1 and we numerically measure classification capacity as $\alpha_c = P/n_c$ where the fraction surpasses 50%; a binary search for the exact value $n_c$ is used to find this transition. The full procedure is described in Supplementary Methods 2.2.

When numerically measuring the capacity of balls with geometric properties derived from the data (as in Fig. 10d) the centers of the balls (and thus their center correlation structure) is taken from the data, as well as the direction of the manifold axes. The number of axes is set by rounding $D_M$ to the nearest integer and manifold radius is $R_M$ in units of the corresponding center norm. Then a specialized method the for measuring linear separability of balls is used[41].

**Generating point-cloud and smooth manifolds.** The pixel-level representation of each point-cloud manifold is generated using samples from a single class from ImageNet data-set[42]. We have chosen $P = 50$ classes (the names and identifiers of the first set of classes used are provided in Supplementary Methods 2.3). For the generation of confidence intervals 5 sets of $P = 50$ classes, sampled with different seeds, were used. The extent of point-clouds can be varied by utilizing the scores assigned to each image by a network trained to classify this data-set, essentially indicating how template-like an image is. Thus we consider here two types of manifolds: (1) "full class" manifolds, where all exemplars from the given class are

used, or (2) "top 10%" manifolds, where just the 10% of the exemplars with large confidence in class-membership, as measured by the score achieved in the soft-max layer, at the node corresponding to the ground-truth class of the exemplar image in ImageNet (a pretrained AlexNet model from PyTorch implementation was used for the score throughout).

The pixel-level representation of each smooth manifold is generated from a single ImageNet image. Only images with an object bounding-box annotation[42] were used; at the base image the object occupied the middle 75% of a $64 \times 64$ image. Manifolds samples are then generated by warping the base image using an affine transformation with either 1 or 2 degrees of freedom. Here we have used 1-d manifolds with horizontal or vertical translation, horizontal or vertical shear; and 2-d manifolds with horizontal and vertical translation or horizontal and vertical shear. The amount of variation in the manifold is controlled for by limiting the maximal displacement of the object corners, thus allowing for generating manifolds with different amount of variability. Manifolds with maximal displacement of up to 16 pixels where used; the resultant amount of variability is quantified by the value of input variability, the amount of variation around manifold center at the pixel layer, in units of the center norm (shown in Fig. 5, Supplementary Figs. 4, 8, measured using Supplementary Eq. (3)). Here $P = 128$ base images were used to generate 1-d manifolds and $P = 64$ to generate 2-d manifolds, both without using images of the same ImageNet class. For the generation of confidence intervals 4 sets of $P = 64$ base images, sampled with different seeds, were used. The number of samples for each of those manifolds is chosen such that capacity would approximately saturate, thus allowing to extrapolate to the case of infinite number of samples (Supplementary Fig. 3).

For both point-cloud and smooth manifolds, representations for all the layers along the different deep hierarchies considered are generated from the pixel-level representation by propagating the images along the hierarchies. Both PyTorch[61] and MatConvNet[62] implementations of the DCNNs were used. At each layer a fixed population of $N = 4096$ neurons was randomly sampled once and used in the analysis, both when numerically calculating capacity and when measuring capacity and manifold properties using the mean-field theory. The first layer of each network is defined as the pixel layer; the last is the feature layer (i.e., before a fully connected operation and a soft-max nonlinearity). Throughout the analysis convolutional and fully connected layers are analyzed after applying local ReLU nonlinearity (unless referring explicitly to the isolated operations as in Fig. 8, Supplementary Fig. 11).

**Image contents**. Images in Figs. 1, 2, 3, and 5 are not the actual ImageNet images used in our experiments. The original images are replaced with images with similar content for display purposes.

**Reporting summary**. Further information on research design is available in the Nature Research Reporting Summary linked to this article.

## Data availability
The datasets generated and analyzed during the current study are available from the corresponding author on reasonable request.

## Code availability
The code used during the current study is publically available in github (see https://github.com/sompolinsky-lab/dnn-object-manifolds).

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

## Acknowledgements

We would like to thank Jim DiCarlo, Yair Weiss, Aharon Azulay, and Joel Dapello for valuable discussions. The work is partially supported by the Gatsby Charitable Foundation, the Swartz Foundation, the National Institutes of Health (Grant No. 1U19NS104653) and the MAFAT Center for Deep Learning. D.D.L. acknowledges support from the NSF, NIH, DOT, ARL, AFOSR, ONR, and DARPA.

## Author contributions

U.C., S.C., D.D.L., and H.S. contributed to the development of the project. U.C. and S.C. performed simulations. U.C., S.C., D.D.L., and H.S. participated in the analysis and interpretation of the data. U.C., S.C., D.D.L., and H.S. wrote the paper.

## Competing interests

The authors declare no competing interests.
