## [Peer Review File · Nature Communications]

Reviewers' Comments:

Reviewer #1:

Remarks to the Author:

This study examines the theoretical underpinning of how the activity of populations of neurons represent information. More specifically, it explores how the properties of the neural activity space that encodes objects (manifolds) change as activity propagates through multilayer networks with different connectivity. Importantly, changes in manifold geometry are related to object classification capacity, which throws new light on the roles played by different layers of deep neural networks. This is a very interesting study and the techniques within this manuscript represent a significant advance in our ability to characterise and understand the separation of manifolds across different regions. The relevance of these new techniques to neuroscience and machine learning make this study of potentially wide impact. However, there are a number of issues that make the current version of the manuscript unsuitable for publication in Nature Communications.

Major issues

- 1) The manuscript is inaccessibly written, which is a pity considering the broad relevance it could have in neuroscience. Given the readership of Nature Communications, the presentation should be suitable for a broader audience than specialists in theoretical neuroscience/machine learning. As the basic concepts are geometrically intuitive, the manuscript could be made significantly more impactful by improving the writing quality, adding more precise definitions, and adding a few more illustrations.
- 2) Several of the claims made are correlational in nature or stated without providing direct evidence. As the authors are aware, there is a correlation vs causation controversy regarding manifolds in neuroscience, so in a computational paper it is critical that the causality of the geometry on separability be well established. One of the strengths of using DNNs as a 'model organism' is control over the network connectivity and the ability to measure the manifold structure at each layer. Each of the main claims in the paper should be backed up by evidence.
- 3) Related to the previous point, it would be informative to analyse further the aspects of the geometry that do and do not affect separability in different layers. While the authors show that R_m , D_m , and centre correlations decrease with layer and increase separability, they do not show other key features of the manifold geometry such as shape and distribution of manifold centres. Do these properties matter? Figure S17 goes some way towards answering this, but is lost in the supplementary information. It deserves fuller treatment in the main text.
- 4) There is a glaring lack of statistics used across the paper. The variability of R_m , D_m , etc. across different manifolds should be depicted and proper statistical tests performed when making comparisons of these quantities.

Specific comments and suggestions:

- 5) The introduction could be made more accessible by splitting it up into a basic introduction defining the problem and describing motivation (no math), and later starting the results with a short section on "previous findings" from their theory - this would comprise the first few paragraphs from pg 2. Following this, a simple figure illustrating object manifolds and separability (like Fig 1 in the authors' PRX paper) would be really useful.
- 6) There is only 1 citation to a neuroscience paper for "It has been hypothesised that the visual hierarchy into linearly separable ones. This approach underlies a number of studies on object representation in the brain ..."

7) Please explain α_c more precisely on pg 2 (ie please explain what “with high probability” means in words). Does this assume randomly chosen subsets of manifolds for the classification?

8) The second paragraph of pg. 2 suddenly dives deep into detail. It is unclear why these technical details are in the middle of the introduction or what we are supposed to take away from these examples. Please move and/or unpack.

9) Figure 1 - please add labels to axes, for the casual reader.

10) Please explain what an anchor point is precisely, in words. In the current description - “each manifold contributes a unique point, called an anchor point” - how is an anchor point different from a support vector and what makes it unique? Relatedly, it would be useful to first briefly describe what the margin and support vectors are in plain words here for the general reader.

11) When first describing AlexNet and VGG-16, it would be useful to have a very basic schematic in Figure 2 to shown pictorially what relu, max pool, convolution, FC mean. This would be really useful for non-specialists especially for understanding the nice results at the end of the paper.

12) The correlation vs. causation issue (general point 2) is illustrated in Figure 2cd. The authors point out the increase in capacity with layer and the ‘concomitant’ decrease in dimensionality and radius. It certainly looks like there is a sharp transition at ~ 7 layers in all of these plots. However, if you look at the results for the full class (not just the top 10%) in figures SI1a, SI3a, SI4a, you still see a sharp increase in α_c in deep layers, but R_m and D_m decrease much more gradually in these cases. Why? Is it that the theory breaks down for shallow layers and for higher variability object manifolds, because overlap and shape become important (which are ignored by the effective D_m/R_m metrics)? If so, that is worth testing and explicitly stating.

13) Figure 2cd (and Figures 5,6)- Shouldn’t there be error bars corresponding to the different object manifolds?

14) The statement “This shuffling destroys any structure of the data, leaving only residual capacity due to the finite number of samples per manifold” should be tested by shuffling the smoothed examples and seeing if the residual capacity decreases.

15) Why is the random labels case worse than before training? Are the before training weights randomised?

16) Smooth manifolds - when calculating α_c , etc. Is each manifold defined by (1) all the transformations of a single image, or (2) all the transformations of all images in the original object category? If (1), then this may not be fully analogous to the true smooth object manifold. Translations of a single image would likely result in manifolds that are much less variable, because they lack the intrinsic variability due to different exemplars from within an object class (eg, there is more variability and probably more structure in the ‘dog’ category than in the ‘head cabbage’ category). Given these considerations, are the conclusions from these smoothed manifolds general when talking about true object manifolds.

17) In the description of figure 4 - ‘Smooth, almost monotonic’ - the curves don’t appear particularly smooth and 4c is not ‘almost monotonic’. How about just ‘increasing’? Also do you have any intuition as to why the increase in 4b is more gradual than in 4c? Figure 4a - please make this figure larger and also include an example of shear transformations. Figure 4d - please define ‘input variability’ in words,

rather than citing an equation in the supplementary information. Also you may want to be reword the term ' α_c increase', as it could be easily conflated with the increase in Figure 3.

18) In relation to the text stating 'the decreased dimension along the deep hierarchies is the main source of the observed increase in capacity from figures 3,4', this would be more convincing if a causal link could be established. Could the authors vary the D_m and R_m of the manifolds in different layers without changing other aspects of the geometry and see how that affects capacity? Alternatively, they could vary the shape of the manifolds without changing D_m or R_m and show that there is no effect on the capacity.

19) Where is that data for the statements 'can also be observed in other measures such as spectral participation ratio' ?

20) 'which can be seen as evidence for the ability of this convolution layer to overcome much of the effects of translation' – This seems too strongly worded. Please consider changing 'evidence for' to 'supports' or 'is consistent with' given that it is a correlational observation.

21) Figure 7 (also Figure 8) – the full class is notably absent, are the results the same? Also why are there so few points in figure 7? Are the results for all manifolds being averaged? It would be useful to show the results for all manifolds to get an idea of the variability, and this would give you enough points to generate a smoothed heatmap for the distribution (pooling layers and network architectures).

22) The sentence 'This analysis highlights how the network architecture consistently reduces manifold correlations at the initial stages of the network and reduces the manifolds' dimension and radius at the final stages' - doesn't agree with Figure 5, which shows D_m and R_m slowly decaying over a broad range of later layers including convolutional layers. Even for smooth manifolds, this description does not match what is shown in Figure 5.

23) Figure 5a VGG-16 smooth manifolds – what explains the pronounced bump? The sentence 'It also explains the non-monotonic behavior of the manifold dimensions in 5a, where dimension increases in sequences of convolutional stages without intermediate pooling' is too strong/general as VGG-16 has several other layers on Conv-Conv-MaxP, and only the first one displays the non-monotonicity.

24) Overall, the manuscript needs to be more inclusive in its citations to the relevant literature. Please include citations for 'Other studies of object representations ...'. The sentence 'To the best of our knowledge this is the first study that highlights the overall decrease of correlations between object manifolds' requires rewording as center correlations are just signal correlations, which have been studied quite extensively – again please acknowledge this literature. 'Decorrelation between neuronal responses' - there is an extensive literature behind this topic, which deserves citation here.

25) Where will the corresponding code be made available?

Reviewer #2:

Remarks to the Author:

Cohen et al., Separability and geometry of object manifolds in deep neural networks

The authors use a recently developed theoretical framework to characterize separability of object manifolds in deep neural networks. The theory allows them to characterize two geometric properties

of manifolds, the radius and dimension, and relate these properties to the capacity of the network to generate representations which allow for arbitrary binary classifications of the manifolds. They analyze several recently developed deep neural networks that are being used for object classification.

There has recently been a strong development of interest in deep neural networks, particularly their application to object recognition. The networks have achieved remarkable performance. However, little is known, theoretically, about how the networks achieve high levels of performance. This manuscript makes substantial progress toward understanding how these networks operate. The theory is detailed, elegant, and exceptionally powerful. The manuscript is reasonably well written for a broad audience. I have a number of comments which could help clarify the presentation and maybe further develop the ideas.

Comments

1. The final paragraph of the introduction ends a bit flat. The authors discuss much of the work that has been done in this space but should add a few sentences at the end to explain how their work moves beyond what has been done.
2. It is often the case that there is little gain in capacity or changes in geometry across the first several convolution layers of the network, followed by a large gain in the last few layers. This is particularly true, for example, for the VGG-16 network in Fig. 3b. This is an interesting finding, which the authors also discuss. Is it possible that several of the convolution layers could be removed without a loss of performance? In the discussion it says, "This does not imply that previous stages are not important." But are they? Is it necessary, as the authors suggest, to have this gradual increase in capacity, and concordant decrease in dimensionality and radius, to setup the large gains shown at the end? Also, is it possible that the first several layers are transforming the data in a way that is not well detected by the theory, but that are important?
3. Related to this, are the effects of layers on dimensionality approximately additive? In other words, if a layer tends to reduce dimensionality or radius by a certain amount, would removing it lead to a change at the end of the network consistent with the amount it contributed? Or do the layers contribute redundantly, and one just needs a sufficient number of layers?
4. What is the difference between Fig. 2c, d and Fig. 5a, b for point cloud manifolds? Should this be the same data?
5. On page 6, the authors state, "Fig. 5a suggests that decreased dimension along the deep hierarchies is the main source of the observed increase in capacity from figures 3, 4." However, isn't it possible to make a more precise statement given the approximation the authors develop using L-2 balls and equations 3-5 of the methods?
6. Can this theory say anything about "adversarial examples" that tend to break deep networks? Do adversarial examples lead to large changes in anchor points for a given manifold?
7. Not sure if the authors can give a bit more intuition about the definitions of equations 2, 3 and 4 in the methods. I realize these are developed in detail in the referenced papers by the authors. But, for example, the sentence "The Gaussian vector T represents the contribution part of the variability in V due to quenched variability in the manifolds' orientation and labels.", is very dense. Anchor points, relevant to equation 3 are also not really explained. These are key to the manuscript. A basic intuition follows from the terms radius and dimensionality, but it would help to link these intuitions to the geometric quantities.

Bruno Averbeck

Reviewer #3:
None

Letter to the editor and response to reviewers

Dear Editor,

We are grateful to the reviewers for their insightful and constructive comments. We appreciate that both reviewers found our study very interesting and of potentially wide impact to both neuroscience and machine learning. We have thoroughly revised the manuscript to address both the concerns regarding “accessibility of presentation” as well as regarding the statistical analysis and causality demonstration. We believe we have addressed all their concerns in the revised manuscript. These revisions have significantly improved the study as well as its presentation. We therefore resubmit our work for publication in Nature Communications.

The main revisions are:

- revised Introduction and a new subsection at the beginning of the Results to better explain the theoretical framework;
- main results were regenerated using different sets of manifolds, and error bars due to manifold sampling added to the appropriate Results figures;
- a new Results subsection that address causality issues with a new figure 10 and additional supplementary figure;
- we have discovered a minor error in our code for evaluating radius and dimensionality of point clouds. This has now been corrected in all relevant figures. Conclusions are unaffected. (The corrected results improved agreement with theory).

Below we respond to the questions and concerns, quoted in full and followed by our response.

Reviewer 1 (R1) major issues:

1) *The manuscript is inaccessibly written, which is a pity considering the broad relevance it could have in neuroscience. Given the readership of Nature Communications, the presentation should be suitable for a broader audience than specialists in theoretical neuroscience/machine learning. As the basic concepts are geometrically intuitive, the manuscript could be made significantly more impactful by improving the writing quality, adding more precise definitions, and adding a few more illustrations.*

5) *The introduction could be made more accessible by splitting it up into a basic introduction defining the problem and describing motivation (no math), and later starting the results with a short section on “previous findings” from their theory - this would comprise the first few paragraphs from pg 2. Following this, a simple figure illustrating object manifolds and separability (like Fig 1 in the authors’ PRX paper) would be really useful.*

We have thoroughly revised the presentation of the theory and the analysis and provided clearer definitions of the various relevant quantities. Following the reviewer’s suggestion, the Introduction is limited to a general background of the problem and our approach and includes a new schematic figure (now figure 1). The main theoretical framework is explained in the new first section of the results called “Geometrical Framework”, and includes also a new figure (now figure 2) that illustrates the notion of Anchor Points.

2) *Several of the claims made are correlational in nature or stated without providing direct evidence. As the authors are aware, there is a correlation vs causation controversy regarding manifolds in neuroscience, so in a computational paper it is critical that the causality of the*

geometry on separability be well established. One of the strengths of using DNNs as a ‘model organism’ is control over the network connectivity and the ability to measure the manifold structure at each layer. Each of the main claims in the paper should be backed up by evidence.

3) Related to the previous point, it would be informative to analyse further the aspects of the geometry that do and do not affect separability in different layers. While the authors show that R_M , D_M , and centre correlations decrease with layer and increase separability, they do not show other key features of the manifold geometry such as shape and distribution of manifold centres. Do these properties matter? Figure S17 goes some way towards answering this, but is lost in the supplementary information. It deserves fuller treatment in the main text.

18) In relation to the text stating ‘the decreased dimension along the deep hierarchies is the main source of the observed increase in capacity from figures 3,4’, this would be more convincing if a causal link could be established. Could the authors vary the D_M and R_M of the manifolds in different layers without changing other aspects of the geometry and see how that affects capacity? Alternatively, they could vary the shape of the manifolds without changing D_M or R_M and show that there is no effect on the capacity.

We thank the reviewer for this comment and have run new extensive computer simulations to demonstrate causality. We have summarized the results in a new section of the Results titled “The effect of manifold perturbations on classification capacity”, summarized in a new figure 10. Briefly, we have done the following manipulations of the geometry:

(1) Scaling- manifolds have been scaled by a global scale factor (relative to their center). This changes the size of the manifold but not its shape. We show that the capacity is monotonically decreasing with this factor, demonstrating the causal effect of manifold sizes on their separability.

(2) We have replaced the original manifolds with balls with the same radius and dimensions as R_M and D_M . Consistent with the theoretical prediction, the capacity is nearly the same as that of the original manifolds, confirming our statement that R_M and D_M and not other salient features of the shape are the dominant factors determining the capacity. In response to Reviewer 2 (below) we have included two additional manipulations.

(3) In calculating the capacity of equivalent balls, we have fixed their radius to be equal to the pixel layer R_M while, as before, their dimension was matched to the D_M of each layer. We show that the enhancement of capacity across the layers is between 80% to 50% of the full capacity (figures 10 and supplementary figure SI7).

(4) on the other hand, fixing the balls’ dimensions to be the D_M of the pixel layer while their radius equaling R_M of each layer, results in a very modest capacity gain across the layers. Taken together these manipulations support our conclusion that the reduction in D_M plays a dominant (but not exclusive) role in capacity gain across the layers. Finally,

(5) We have quantified the effect of correlations by randomizing the centers (as was done in the original manuscript). Results are shown in supplementary figure SI10 and summarized in the main text.

4) There is a glaring lack of statistics used across the paper. The variability of R_M , D_M , etc. across different manifolds should be depicted and proper statistical tests performed when making comparisons of these quantities.

We have performed extensive simulations to address this issue. We have repeated our main results over new samplings of manifolds and calculated the generated variability of capacity, radius and dimensions. Figures 3-7 displays the results with 95% confidence intervals. As can be seen, the variations due to the sampling of manifolds are substantially smaller than the systematic change across layers and does not change our conclusions.

Reviewer 1 (R1) specific comments and suggestions:

6) *There is only 1 citation to a neuroscience paper for “It has been hypothesised that the visual hierarchy into linearly separable ones. This approach underlies a number of studies on object representation in the brain ...”*

Corrected.

7) *Please explain α more precisely on pg 2 (ie please explain what “with high probability” means in words). Does this assume randomly chosen subsets of manifolds for the classification?*

Explained.

8) *The second paragraph of pg. 2 suddenly dives deep into detail. It is unclear why these technical details are in the middle of the introduction or what we are supposed to take away from these examples. Please move and/or unpack.*

Explained and moved into first subsection of Results.

9) *Figure 1 - please add labels to axes, for the casual reader.*

Added to new figures 1 and 2.

10) *Please explain what an anchor point is precisely, in words. In the current description - “each manifold contributes a unique point, called an anchor point” - how is an anchor point different from a support vector and what makes it unique? Relatedly, it would be useful to first briefly describe what the margin and support vectors are in plain words here for the general reader.*

Done. For the sake of ‘general audience’ we have removed explicit reference throughout to margin. The reason is that as we anyway work near capacity, in which the separating plane is essentially unique and characterized by the support points or anchor points.

11) *When first describing AlexNet and VGG-16, it would be useful to have a very basic schematic in Figure 2 to shown pictorially what relu, max pool, convolution, FC mean. This would be really useful for non-specialists especially for understanding the nice results at the end of the paper.*

Done, added to figure 3.

12) *The correlation vs. causation issue (general point 2) is illustrated in Figure 2cd. The authors point out the increase in capacity with layer and the ‘concomitant’ decrease in dimensionality and radius. It certainly looks like there is a sharp transition at 7 layers in all of these plots. However, if you look at the results for the full class (not just the top 10%) in figures SI1a, SI3a, SI4a, you still see a sharp increase in α in deep layers, but R_m and D_m decrease much more gradually in these cases. Why? Is it that the theory breaks down for shallow layers and for higher variability object manifolds, because overlap and shape become important (which are ignored by the effective D_m/R_m metrics)? If so, that is worth testing and explicitly stating.*

We believe the theory holds also in this case. To demonstrate it, we have included in figure 10b comparison between balls and original manifolds (with the same R_M and D_M (see above) for both ”top 10% and for ”full class” point cloud manifolds. The agreement is excellent in both, implying that also for the manifolds depicted in (old) figures SI1a, SI3a, SI4a, the relatively shallow capacity curve in the first stages, is due to a floor effect: capacity remains close to the random shuffled value, until there is a significant reduction in dimensionality.

13) *Figure 2cd (and Figures 5,6)- Shouldn’t there be error bars corresponding to the different object manifolds?*

Added.

14) *The statement “This shuffling destroys any structure of the data, leaving only residual capacity due to the finite number of samples per manifold” should be tested by shuffling the smoothed examples and seeing if the residual capacity decreases.*

Smooth manifolds are densely sampled and essentially consist of infinite number of points, hence shuffling results in practically zero capacity $2/M \approx 0$.

15) *Why is the random labels case worse than before training? Are the before training weights randomised?*

Yes, the weights before training are randomized. However, network has the same architectural features as that after training (e.g., convolutions, pooling, nonlinearity). These effects improve the geometry and separability of manifold to some degree. On the other hand, the random labels, destroy any structure in the manifolds, reducing the problem to that of classifying random points, hence the network has virtually no impact (the shuffled points remain randomly dispersed in all layers). In fact, we now have done a theoretical analysis of the expected properties of random points (not only for the capacity- $2/M$ but also for R_M and D_M) and show in figure 4 that the flat values of the shuffled manifolds agree very well with these predictions.

16) *Smooth manifolds - when calculating α , etc. Is each manifold defined by (1) all the transformations of a single image, or (2) all the transformations of all images in the original object category? If (1), then this may not be fully analogous to the true smooth object manifold. Translations of a single image would likely result in manifolds that are much less variable, because they lack the intrinsic variability due to different exemplars from within an object class (eg, there is more variability and probably more structure in the ‘dog’ category than in the ‘head cabbage’ category). Given these considerations, are the conclusions from these smoothed manifolds general when talking about true object manifolds.*

The smooth manifolds are defined as representing template images (which underwent the said distortions). They are not supposed to represent the full extent of variability of object manifolds. Estimating the full variability of object manifolds is a difficult problem beyond the scope of this work. The utility of the smooth manifold is the fact that they nicely model the effect of a continuous smooth variability induced by physical factors such as translation, shear etc. In contrast to point clouds, their analysis is not limited by the finite number of stimuli, and demonstrate the utility of our methods also in manifolds with infinite number of points. Finally, as we now note in the text, such manifolds arise naturally in neuroscience experiments which often manipulate stimuli by changing one or two physical variables.

17) *In the description of figure 4 - ‘Smooth, almost monotonic’ - the curves don’t appear particularly smooth and 4c is not ‘almost monotonic’. How about just ‘increasing’? Also do you have any intuition as to why the increase in 4b is more gradual than in 4c? Figure 4a - please make this figure larger and also include an example of shear transformations. Figure 4d - please define ‘input variability’ in words, rather than citing an equation in the supplementary information. Also you may want to be reword the term ‘ α increase’, as it could be easily conflated with the increase in Figure 3.*

Amended.

19) *Where is that data for the statements ‘can also be observed in other measures such as spectral participation ratio’?*

Added in supplementary figure SI5 and referred to in the main text.

20) *‘which can be seen as evidence for the ability of this convolution layer to overcome much of the effects of translation’ - This seems too strongly worded. Please consider changing ‘evidence for’ to ‘supports’ or ‘is consistent with’ given that it is a correlational observation.*

Modified.

21) *Figure 7 (also Figure 8) - the full class is notably absent, are the results the same? Also why are there so few points in figure 7? Are the results for all manifolds being averaged?*

It would be useful to show the results for all manifolds to get an idea of the variability, and this would give you enough points to generate a smoothed heatmap for the distribution (pooling layers and network architectures).

The full class results are incorporated in (old) figure 7. The small number of points in the FC panels results from the fact that there are only few such layers in each network. With regards to the reviewer’s suggestion, we prefer not to include individual manifolds as the x axis denotes correlations which are not defined for individual manifolds. With regards to (old) figure 8, this figure is based on numerical simulations of classification by a hyperplane of the real data. Given the low (absolute) capacity of the full manifolds in large parts of the network it is infeasible to simulate such classification. Thus, for full manifolds we only analyze the theoretical estimation of capacity.

22) The sentence ‘This analysis highlights how the network architecture consistently reduces manifold correlations at the initial stages of the network and reduces the manifolds’ dimension and radius at the final stages’ - doesn’t agree with Figure 5, which shows D_m and R_m slowly decaying over a broad range of later layers including convolutional layers. Even for smooth manifolds, this description does not match what is shown in Figure 5.

Clarified.

23) Figure 5a VGG-16 smooth manifolds – what explains the pronounced bump? The sentence ‘It also explains the non-monotonic behavior of the manifold dimensions in 5a, where dimension increases in sequences of convolutional stages without intermediate pooling’ is too strong/general as VGG-16 has several other layers on Conv-Conv-MaxP, and only the first one displays the non-monotonicity.

Clarified.

24) Overall, the manuscript needs to be more inclusive in its citations to the relevant literature. Please include citations for ‘Other studies of object representations . . .’. The sentence ‘To the best of our knowledge this is the first study that highlights the overall decrease of correlations between object manifolds’ requires rewording as center correlations are just signal correlations, which have been studied quite extensively – again please acknowledge this literature. ‘Decorrelation between neuronal responses’ - there is an extensive literature behind this topic, which deserves citation here.

We have reworded the sentence and added additional citations, both overall, and specifically for other studies of object representations and decorrelation between neuronal responses.

25) Where will the corresponding code be made available?

The code will be made publically available in github. We now mention this in the code availability section.

Reviewer 2 (R2) comments:

1. The final paragraph of the introduction ends a bit flat. The authors discuss much of the work that has been done in this space but should add a few sentences at the end to explain how their work moves beyond what has been done.

We believe the modified Introduction addresses this point.

2. It is often the case that there is little gain in capacity or changes in geometry across the first several convolution layers of the network, followed by a large gain in the last few layers. This is particularly true, for example, for the VGG-16 network in Fig. 3b. This is an interesting finding, which the authors also discuss. Is it possible that several of the convolution layers could be removed without a loss of performance? In the discussion it says, “This does not imply that previous stages are not important.” But are they? Is it necessary, as the authors suggest, to have this gradual increase in capacity, and concordant decrease in dimensionality and radius,

to setup the large gains shown at the end? Also, is it possible that the first several layers are transforming the data in a way that is not well detected by the theory, but that are important?

3. Related to this, are the effects of layers on dimensionality approximately additive? In other words, if a layer tends to reduce dimensionality or radius by a certain amount, would removing it lead to a change at the end of the network consistent with the amount it contributed? Or do the layers contribute redundantly, and one just needs a sufficient number of layers?

It is difficult to conclusively test this issue as it is unclear how to manipulate the network to 'shorten it'. Often, this would require adjustment of layer sizes, or in general retraining of the modified networks. However, we have now added in supplementary figure SI2 the capacity and geometry changes in three ResNet networks of different lengths overlaid on each other. The figure nicely illustrates that the properties of initial stages of all the networks are similar and differ mainly at the respective last sections (following a substantial downsampling). Thus it suggests that the difference in the final performance is related to the long period of incremental improvements in the deeper architectures. However, we do not think we have a conclusive way to test questions like additivity of layer contributions.

4. What is the difference between Fig. 2c, d and Fig. 5a, b for point cloud manifolds? Should this be the same data?

Figure 5 repeats the results of (old) figure 2c,d but compares with other manifolds. Figure captions changed to state this is the case.

5. On page 6, the authors state, "Fig. 5a suggests that decreased dimension along the deep hierarchies is the main source of the observed increase in capacity from figures 3, 4." However, isn't it possible to make a more precise statement given the approximation the authors develop using $L-2$ balls and equations 3-5 of the methods?

As described above, we now quantify it in the new figure 10b and supplementary figure SI7.

6. Can this theory say anything about "adversarial examples" that tend to break deep networks? Do adversarial examples lead to large changes in anchor points for a given manifold?

It is a tantalizing possibility which we pursue in ongoing research. We feel it is too premature to speculate about it in this paper.

7. Not sure if the authors can give a bit more intuition about the definitions of equations 2, 3 and 4 in the methods. I realize these are developed in detail in the referenced papers by the authors. But, for example, the sentence "The Gaussian vector T represents the contribution part of the variability in V due to quenched variability in the manifolds' orientation and labels.", is very dense. Anchor points, relevant to equation 3 are also not really explained. These are key to the manuscript. A basic intuition follows from the terms radius and dimensionality, but it would help to link these intuitions to the geometric quantities.

Following the reviewer's suggestion we have expanded this section of the Methods to provide such intuition.

Reviewers' Comments:

Reviewer #1:

Remarks to the Author:

The authors have done a good job in revising the manuscript and have addressed the weaknesses in the initial submission. It is now much more accessible and the new simulations provide much more compelling evidence for the theory.

Here are a few minor remaining points the authors might want to consider when preparing a final version.

Figure 1 would be clearer if the red and blue manifolds were less separated in the pixels/retina and intermediate layers. This would better get across the idea that the manifolds are being disentangled and shrunk by the network.

P4 "shorter network" -> "shallower network".

There is an error in figure 10 legend when describing cyan lines - should be for panel b not c.

Reviewer #2:

Remarks to the Author:

The authors have done a nice job of addressing my comments. I think Fig. SI2 is quite interesting. This question does not have to be addressed here. But it seems like an interesting empirical question to explore. Is it possible that with longer training more of the early layers would pick-up additional ability to reduce radius or dimension? I'm wondering if the networks classification abilities are working backwards from the outputs, since the effect of any synapse on the output will be decreased as one moves further from the output of the network.

Bruno Averbeck

Response to reviewers

We are grateful to both reviewers for their insightful and constructive comments throughout the entire peer-review process. Your feedback and suggestions have considerably improved the manuscript and we appreciate your thoughtful comments and evident efforts.

Reviewer #1

Figure 1 would be clearer if the red and blue manifolds were less separated in the pixels/retina and intermediate layers. This would better get across the idea that the manifolds are being disentangled and shrunk by the network.

Done.

P4 "shorter network" -> "shallower network".

Done.

There is an error in figure 10 legend when describing cyan lines- should be for panel b not c.

Done.

Reviewer #2

I think Fig. SI2 is quite interesting. This question does not have to be addressed here. But it seems like an interesting empirical question to explore. Is it possible that with longer training more of the early layers would pick-up additional ability to reduce radius or dimension? I'm wondering if the networks classification abilities are working backwards from the outputs, since the effect of any synapse on the output will be decreased as one moves further from the output of the network.

We thank for the reviewer for this suggestion. However, it is beyond the scope of current paper to systematically study the dynamics of manifold geometry during learning. Indeed, the behavior at the initial layers during learning may be non-trivial (e.g. is it quickly saturating or slowly improving, and is it aligned with the behavior at the final layers) and these questions are the subject of future research.